# Interleukin-33 Induces Neutrophil Extracellular Trap (NET) Formation and Macrophage Necroptosis via Enhancing Oxidative Stress and Secretion of Proatherogenic Factors in Advanced Atherosclerosis

**DOI:** 10.3390/antiox11122343

**Published:** 2022-11-26

**Authors:** Manoj Kumar Tembhre, Mukesh Kumar Sriwastva, Milind Padmakar Hote, Shikha Srivastava, Priyanka Solanki, Shafaque Imran, Ramakrishnan Lakshmy, Alpana Sharma, Kailash Jaiswal, Ashish Datt Upadhyay

**Affiliations:** 1Department of Cardiac Biochemistry, All India Institute of Medical Sciences (AIIMS), New Delhi 110029, India; 2Brown Cancer Center, University of Louisville, Louisville, KY 40202, USA; 3Department of Cardiothoracic & Vascular Surgery, C. T. Centre, AIIMS, New Delhi 110029, India; 4Department of Microbiology & Immunology, University of Louisville, Louisville, KY 40202, USA; 5Department of Biochemistry, AIIMS, New Delhi 110029, India; 6Clinical Research Unit (CRU), Biostatistics, AIIMS, New Delhi 110029, India

**Keywords:** Interleukin-33, NETs, necroptosis, advanced atherosclerosis, myeloperoxidases (MPO), macrophages

## Abstract

Interleukin-33 (IL-33) acts as an ‘alarmin’, and its role has been demonstrated in driving immune regulation and inflammation in many human diseases. However, the precise mechanism of action of IL-33 in regulating neutrophil and macrophage functioning is not defined in advanced atherosclerosis (aAT) patients. Further, the role of IL-33 in neutrophil extracellular trap (NET) formation in aAT and its consequent effect on macrophage function is not known. In the present study, we recruited *n* = 52 aAT patients and *n* = 52 control subjects. The neutrophils were isolated from both groups via ficoll/percoll-based density gradient centrifugation. The effect of IL-33 on the NET formation ability of the neutrophils was determined in both groups. Monocytes, isolated via a positive selection method, were used to differentiate them into macrophages from each of the study subjects and were challenged by IL-33-primed NETs, followed by the measurement of oxidative stress by calorimetric assay and the expression of the proinflammatory molecules by quantitative PCR (qPCR). Transcript and protein expression was determined by qPCR and immunofluorescence/ELISA, respectively. The increased expression of IL-33R (ST-2) was observed in the neutrophils, along with an increased serum concentration of IL-33 in aAT compared to the controls. IL-33 exacerbates NET formation via specifically upregulating CD16 expression in aAT. IL-33-primed NETs/neutrophils increased the cellular oxidative stress levels in the macrophages, leading to enhanced macrophage necroptosis and the release of atherogenic factors and matrix metalloproteinases (MMPs) in aAT compared to the controls. These findings suggested a pathogenic effect of the IL-33/ST-2 pathway in aAT patients by exacerbating NET formation and macrophage necroptosis, thereby facilitating the release of inflammatory factors and the release of MMPs that may be critical for the destabilization/rupture of atherosclerotic plaques in aAT. Targeting the IL-33/ST-2-NETs axis may be a promising therapeutic target for preventing plaque instability/rupture and its adverse complications in aAT.

## 1. Introduction

Cardiovascular disease (CVD) is considered to be the leading cause of morbidity and mortality globally, and the culprit cause is atherosclerosis [1]. Atherosclerosis is characterized by dysfunction in the endothelial cells, lipid deposition and oxidation, and inflammatory cell infiltration, leading to plaque formation followed by arterial wall thickening and a narrowing of the lumens in the arteries [2]. These features of atherosclerosis cause clinical complications, such as ischemic heart disease, stroke, myocardial infarction, and peripheral arterial disease [1,2]. The most abundant immune cells are neutrophils, and their role has been implicated in the pathogenesis of various autoimmune and inflammatory diseases, including atherosclerosis [3,4,5]. Neutrophils are generally known as the primary immune cells that are critical in host defense against pathogenic microorganisms via microbial uptake and phagocytosis [6]. The neutrophil extracellular traps (NETs) or NETosis is a recently identified cell death mechanism (distinct from apoptosis and necrosis), and NETs are formed in response to pathogens and under the influence of myriad of inflammatory factors [6,7]. However, the phenomenon of extracellular trap formation is not limited to neutrophils, and recent studies have demonstrated a similar mechanism in other immune cells (macrophages, eosinophils, mast cells, etc.) [8,9,10].

The NET armamentarium mainly includes decondensed chromatin fibers complexed with histone protein, along with various inflammatory molecules (HMGB1, cathepsin, myeloperoxidase, etc.) and antimicrobial peptides (cathelicidin, calgranulin, defensins, pentraxin, etc.) [6,7]. In recent years, substantial studies reported the involvement of NETs in the development and progression of atherosclerosis [11,12,13]. Warnatsch et al. demonstrated that NETs induced by cholesterol crystals have the potential to activate the inflammatory Th17 cells, priming the macrophages to secrete the proinflammatory cytokines, leading to the progression of atherosclerotic plaques [12]. NETs are also reported to activate endothelial cells, endothelial damage, platelet activation, monocyte adhesion, and foam cell formation, which are considered to be characteristic features of atherosclerosis [14,15,16,17].

Further, interleukin (IL)-33 (IL-1 family cytokine) has recently been identified as an endogenously produced molecule called “alarmins” that acts as a danger signal and possesses potent inflammatory properties [18,19]. When considering its alarmin function, it is often referred to as a damage-associated molecular pattern (DAMP) molecule and is released after tissue or cell injury as a result of necrosis [19,20]. Expression of IL-33 is reported in various immune (eosinophils, dendritic cells, macrophage, mast cells, T cells, and innate lymphoid cells) epithelial and endothelial cells, and its secretion is stimulated by infectious and inflammatory conditions [21,22,23,24,25]. However, its role has also been suggested in augmenting sterile inflammation [20], where pattern recognition receptors (PRRs) and pathogen-associated molecular patterns (PAMPs) are activated by non-microbial signals, and sterile inflammation is considered as one of the causative factors in the atherosclerosis etiology [26,27]. IL-33 exerts its function after engagement with its receptors, i.e., IL-33R (ST-2) and sST-2 (membrane-bound and soluble isoforms), and the full-length IL-33 is the biologically active form [21]. The role of IL-33 has been implicated in various cardiovascular diseases, including atherosclerosis [24]. An elevated concentration of serum IL-33 and ST2, along with the increased expression of IL-33 and IL-33 receptors (IL-33R), were reported in vulnerable plaques, which, in turn, corroborate with a degree of infiltration of inflammatory cells (in the plaques) [28]. However, contradictory studies were also reported where its protective role had been suggested in atherosclerosis [29].

Macrophages are predominant immune cells that are involved in atherosclerotic lesion development and progression [30]. The stability of advanced plaques is associated with macrophage cell death (via apoptosis, necrosis, etc.), which is considered responsible for the formation of necrotic cores and the destabilization of plaques in advanced atherosclerosis (aAT) [30,31]. The role of necroptosis has gained wide attention, and its role has been suggested in various cardiovascular diseases, including atherosclerosis [32,33]. Unlike necrosis, necroptosis occurs in a programmed manner, and it is regulated by RIPK (receptor-interacting protein kinases)-1, RIPK3, and MLKL (mixed-lineage kinase and domain-like pseudokinase). Necroptosis is triggered by a variety of stimuli, including cytokines and other endogenous factors [32], but the role of IL-33 and NETs in triggering the necroptosis in the macrophages of aAT is not defined. Further, the role of IL-33 in NET induction in advanced atherosclerosis is also not understood. Therefore, the present study aimed to define the role IL-33 in augmenting NETs and the effect of IL-33-primed NETs in macrophage necroptosis induction, followed by an investigation of the status of oxidative stress and the release of atherogenic inflammatory molecules and matrix metalloproteinase (MMPs) from macrophages in aAT.

## 2. Materials and Methods

### 2.1. Patient and Control Groups

A total of 52 (*n* = 43 males and *n* = 9 females) advanced atherosclerosis (≥70% stenosis) patients were referred for coronary artery bypass grafting (CABG) after complete clinical evaluation, and 52 (*n* = 42 males and *n* = 10 females) controls were recruited in the study (Table 1). Blood samples were collected from the patient and control groups. Additionally, we collected *n* = 18 carotid endarterectomy plaques from patients who underwent CABG surgery. None of the study subjects (in both groups) were treated with any systemic corticosteroids or other immunosuppressive therapy, preceding one month before blood collection. The control subjects did not have any inflammatory, autoimmune, or infectious disorders and had not taken any medication (topical or oral) in the preceding (one) month. Patients and controls having any past or current history of smoking, alcohol, and drug usage were excluded. The study was approved by the institutional ethics committee, and written informed consent was obtained from each of the study subjects.

### 2.2. Neutrophil Extracellular Trap Formation

Neutrophils were isolated from all the study subjects in the patient and control groups by density gradient centrifugation using Histopaque 1119 (Sigma-Aldrich, St. Louis, MO, USA) and a percoll (Sigma-Aldrich, St. Louis, MO, USA) gradient, as described by Brinkman et al. [34]. Isolated neutrophils were plated onto 24-well culture plates in RPMI media, supplemented with 2% human serum albumin, and were allowed for adherence for one hour, followed by treatment with rIL-33 (100 ng/mL; Sigma-Aldrich, St. Louis, MO, USA) for 4 h in a CO_2_ incubator at 37 °C. For the immunofluorescence-related experiments, the cells were cultured on sterile 13 mm round glass cover slip.

### 2.3. Monocyte Isolation, Macrophage Culture and Stimulation

The peripheral blood mononuclear cells (PBMCs) were isolated by density gradientcentrifugation using Histopaque-1077 (Sigma-Aldrich, St. Louis, MO, USA), followed by monocytes enrichment by positive selection using EasySep™ Human Monocyte Isolation Kit (Stem Cell Technologies Inc., Vancouver, BC, Canada) as per the manufacturer’s instructions. Briefly, 80–100 million PBMCs were incubated for 10 min with a cocktail of selection antibody (100µL/mL), followed by 5 min incubation with RapidSpheres (100 µL/mL, StemCell Technologies Inc., Vancouver, BC, Canada)). The cells were placed in “The Big Easy”EasySep™ Magnet (StemCell Technologies Inc., Vancouver, BC, Canada) for 3–4min; the supernatant (negative for monocytes) was removed, and the monocytes resuspended in buffer (PBS with 1mM EDTA and 2% fetal calf serum). Isolated monocytes (2 × 10^6^) were differentiated into macrophages by culturing them in differentiation media, i.e., RPMI-1640 supplemented with 10% FCS (fetal calf serum), 1× antibiotic-antimycotic (penicillin, streptomycin, and amphotericin-B) solution (Himedia Laboratories, Mumbai, India), 10 ng/mLM-CSF (macrophage colony-stimulating factor) (Sigma-Aldrich, St. Louis, MO, USA), and 1 ng/mLGM-CSF granulocyte-macrophage colony-stimulating factor (Sigma-Aldrich, St. Louis, MO, USA) for 4–5 days. Macrophages were co-cultured with IL-33 primed NETs (neutrophils) and 50% NETs supernatant for 24 h.

### 2.4. Quantitative Polymerase Chain Reaction (qPCR)

The qPCR was performed as described previously [35]. Briefly, the total RNA was isolated using TRIzol^TM^ (Thermo Fisher Scientific, Carlsbad, CA, USA) from cells (neutrophils/macrophages), according to the manufacturer’s instructions. RNase free kit (ThermoFisher Scientific, Carlsbad, CA, USA) with DNase I was used to remove the contaminating DNA. Complementary DNA (cDNA) was synthesized using the RevertAid first strand cDNA synthesis kit (Thermo Fisher Scientific, Carlsbad, CA, USA). Quantitative polymerase chain reaction (PCR) was performed using the Maxima SYBR Green qPCR Master Mix (Thermo Fisher Scientific, Carlsbad, CA, USA) in CFX96 Real-time PCR System (BioRad, Hercules, CA, USA) as per the MIQE guidelines. The PCR settings used were initial denaturation at 95 °C for 10 min, and 40 cycles of 15 s of denaturation at 95 °C, and 30s of primer annealing (at optimized temperature), and extension at 72 °C. Samples were run in triplicates (average Ct values was used for analysis), and β-actin served as the internal control for normalization. Primers were purchased from Sigma-Aldrich. The genes and their primer pairs were listed in Table 2. Gene expression data were expressed as 2^−ΔCt^ for the patient and control groups.

### 2.5. Oxidative Stress Estimation

The oxidative stress was analyzed by estimating the concentration of 8-hydroxy 2 deoxyguanosine (8-OHdG) (Abcam, Cambridge, UK, ab201734), malondialdehyde (MDA)/lipid peroxidation (Abcam, Cambridge, UK, ab118970), GSH+GSSG (total glutathione)/GSH (reduced glutathione) (Abcam, Cambridge, UK, ab239709), superoxide dismutase (SOD) activity (Elabscience Biotechnology Inc., Houston, TX, USA, E-BC-K019-M) and Catalase activity (Elabscience Biotechnology Inc., Houston, TX, USA, E-BC-K031-M) in the cultured macrophages of patients and control groups as explained above. All the steps were performed as per the manufacturer’s instruction, and colorimetric detection method was used in all assays.

### 2.6. Immunofluorescence

For immunofluorescence study, the macrophages were cultured on 13 mm coverslips and treated as mentioned above, followed by fixation in 4% paraformaldehyde. Plaque samples were fixed in 4% paraformaldehyde, followed by incubation in decalcifying solution (Sigma Aldrich, St. Louis, MO, USA), washed and dehydrated under an increasing gradient of alcohol, and embedded in paraffin. Consecutive sections (5 μm thickness) were cut and transferred on poly-L-lysine coated slides (Sigma Aldrich, St. Louis, MO, USA). Sections were deparaffinized and incubated in an antigen retrieval buffer (sodium citrate, pH 6). Fixed cells and plaque sections were washed and permeabilized (only for intracellular markers) using 0.2% Triton (Bio-Rad, Hercules, CA, USA), and blocked in 5% normal goat serum (Abcam, Cambridge, UK). After blocking, the cells were incubated with primary antibodies, polyclonal goat anti-myeloperoxidae (MPO) (1:200, R & D Systems), rabbit anti-CD16 (1:200, Abcam, Cambridge, UK), Rabbit polyclonal IL-33R (ST2) (1:100, Abcam, Cambridge, UK), rabbit anti- RIPK-1(1:200, Abcam, Cambridge, UK), rabbit anti-RIPK-3 (1:200, Abcam, Cambridge, UK) and, rabbit anti-MLKL (phospho S358) (1:200, Abcam, Cambridge, UK) overnight at 4 °C. Cells were washed with 0.025% Tween-20 (in 1XPBS), followed by incubation with corresponding secondary antibodies, i.e., FITC-conjugated rabbit anti-goat (1:500, Abcam, Cambridge, UK), FITC-conjugated goat anti-rabbit (1:500, Abcam, Cambridge, UK), and TRITC-conjugated goat anti-rabbit (1:500, Abcam, Cambridge, UK) for 45 min at room temperature. Cells were washed, and the nuclei were counterstained with 4′,6-diamidino-2-phenylindole (DAPI) (Sigma Aldrich, St. Louis, MO, USA). Cells and plaque sections were mounted on glass slides with Vectashield antifade mounting medium (Vector Laboratories, Newark, CA, USA), and images were acquired using a fluorescence microscope (20×/40× magnification) with NIS-Elements F3.2 software (Nikon, Tokyo, Japan). The mean fluorescence intensity (MFI) was quantified using ImageJ software (NIH, Bethesda, Maryland, USA) for each marker. The intensity threshold levels of the background and signal were determined for at least 5 different fields with the highest signal intensity using the threshold tool. The average threshold values were obtained, and the threshold settings were applied to all images. Then mean fluorescent intensity (MFI) was calculated via the measuring tool for the background and signals, followed by subtracting the MFI of the background from that of the signal. The resulting values of the 5 different fields (in triplicates) were then averaged (normalized with control to obtain fold change of pixel intensities), and the same was used for statistical analysis.

### 2.7. Enzyme Linked Immunosorbent Assay (ELISA)

Protein quantification was performed by ELISA for MPO (sensitivity = <10 pg/mL, Abcam, Cambridge, UK), and IL-33R (ST-2) (sensitivity = 22 pg/mL, Abcam, Cambridge, UK) (in plaque homogenates); RIPK1 (sensitivity = 0.063 ng/mL, ELK Biotechnology, Wuhan, China), RIPK3 (sensitivity = 0.122 ng/mL, ELK Biotechnology), and MLKL (sensitivity = 0.112 ng/mL, ELK Biotechnology) (in treated macrophage cell lysates). All the steps were performed as per the manufacturer’s instructions, and measurements of optical densities were determined at 450nm using microplate reader (BioRad, Hercules, CA, USA). The concentrations were expressed as pg (picogram)/mL(milliliter) or ng (nanogram)/mL.

### 2.8. Statistical Analysis

All statistical analyses were performed, and the graphs were generated using STATA software version 14 (StataCorp LP, College Station, TX, USA) and Graph Pad Prism 5 (GraphPad Software, Inc., San Diego, CA, USA). In the present study, the nonparametric datasets (qPCR, calorimetric assays, and ELISA data) were analyzed using Mann–Whitney U testing (represented as a boxplot graph). The immunofluorescence data (except for Figure 1c, where Mann–Whitney U testing was used) was analyzed using the student’s t test as the data were parametric. *p* value of <0.05 was set as significant. We also compared the data for effect size analysis (i.e., Cohen’s ‘d’ effect size) (95% class interval) for all the datasets, and the same was presented in Appendix A.

## 3. Results

The role of IL-33 has been reported in atherosclerosis, but the precise mechanism is not known, particularly its effect on neutrophil and macrophage function. In the present study, we observed significant (*p* < 0.05) increased expression (transcript and proteins) of IL-33 receptor (IL-33R or IL1RL1 or ST-2) in the neutrophils of the aAT patients compared to the controls (Figure 1a–c). Myeloperoxidase (MPO) is abundantly expressed in neutrophils, and MPO, along with IL-33R, was found to be immunolocalized by immunofluorescence in the endarterectomy plaques of aAT patients suggesting the role IL-33 in the pathogenesis of aAT (Figure 1d,f). We also measured the concentration of MPO and IL-33R in the plaque homogenates by ELISA; the corresponding protein was detected in all the studied plaque samples, but no significant correlation was observed between MPO and IL-33R (*r* = 0.292, *p* = 0.239). Further, we found a significant increase in concentration for the IL-33 in the serum of the aAT patients compared to the controls (*p* < 0.05) (Figure 1e). Next, we determined the effect of rhIL-33 on NET formation potential in the isolated neutrophils of both the groups, where we found a significant increase (*p* < 0.05) in the percentage of NETotic cells in aAT compared to the controls (Figure 2). Furthermore, we found that the enhanced NET formation was mainly associated with CD16+MPO+ neutrophils (Figure 2a) via the upregulation of CD16 expression in the aAT group compared to the controls.

Furthermore, when isolated macrophages were cocultured with IL-33-primed neutrophils/NETs, increased macrophage necroptosis was evident by the significant increased (*p* < 0.05) expression level (protein and mRNA) of RIPK-1, RIPK-3, and MLKL (Figure 3a–d). Since the exacerbation of NETs is associated with an increase in the magnitude of cellular oxidative stress, we estimated the status of oxidative stress in the macrophages treated with IL-33-primed neutrophils/NETs using various markers of oxidative stress. The concentration of8-OHdG and malondialdehyde (MDA) (lipid peroxidation) were significantly higher(*p* < 0.05),and an increase (*p* < 0.05) in the ratio of GSH+GSSG (total glutathione)/GSH (reduced glutathione) was reported in the treated macrophages of aAT compared to the controls (Figure 4). Additionally, the activity of antioxidant enzymes, i.e., superoxide dismutase (SOD) activity and catalase activity, were found to be lowered in NETs-treated macrophages of the aAT group compared to the controls (Figure 4), which is indicative of redox imbalance.

In order to further understand the IL-33-mediated neutrophil-macrophage cross-talk, we determined the transcript levels of the proatherogenic markers in the treated (IL-33 primed neutrophils/NETs) macrophages; we found significantly increased (*p* < 0.05) expression of all the studied markers, i.e., *IFN* (interferon)*-γ*, *TNF* (tumor necrosis factor)*-α*, *IL-1β*, *IL-6*, *IL-12B*, *IL-18*, *IL-23*, *IL-33*, *MCP* (monocyte chemoattractant protein-1)*-1/CCL-2,* and *GM-CSF*(Granulocyte-macrophage colony-stimulating factor), except for *IL-10*, which was not statistically significant between the groups (*p* > 0.05) (Figure 5). The release of inflammatory factors by macrophages was known to be associated with inflammasome activation, and in the present study, we found increased expression of *IL-1β* and *IL-18* (Figure 5), which were indicative of inflammasome activation. We next investigated the expression of common inflammasomes: *NLRP1, NLRP3, NLRC4,* and *AIM2* (Figure 6). No significant difference was observed in the transcript expression of the studied inflammasome markers, except for *NLRP3* expression, which was found to be significantly higher in the treated macrophages of the patients when compared to controls, suggesting the *NLRP3*-mediated activation of inflammasome. Furthermore, we investigated the mRNA expression levels of the tissue inhibitors of metalloproteinases (TIMPs), i.e., *TIMP1, TIMP2, TIMP3,* and *TIMP4*, where *TIMP3* (*p* < 0.05) was found to be significantly decreased in the treated macrophages of the patients when compared to the controls; a similar trend was observed for *TIMP1,* but the difference was not statistically significant (*p* = 0.061) (Figure 6). The role of the MMPs secreted by macrophages was known to be involved in plaque destabilization, along with inflammatory cytokines; therefore, we extended our study to investigate the transcript expression of selective MMPs, i.e., *MMP-1, MMP-2, MMP-3, MMP-7, MMP-9, MMP-12, MMP-13*, and *MMP-14*. We found increased expression of *MMP-3, MMP-9,* and *MMP-12* (*p* < 0.05), but no significant difference was observed for *MMP-1, MMP-2, MMP-7, MMP-12*, and *MMP-14 (p* > 0.05) (Figure 6).

## 4. Discussion

In the present study, we identified a novel mechanism mediated by IL-33 whereby it binds to its membrane-bound receptor, IL-33R, on neutrophils and triggers a potential NET formation response by up-regulating CD16 expression. The IL-33-primed NETs further induced macrophage activation via NLRP3 inflammasome, facilitating the release of atherogenic inflammatory mediators and MMPs. All these events eventually increased the cellular oxidative stress in the macrophages, leading to macrophage necroptosis. The involvement of the IL-33/ST-2 axis has been reported in various cardiovascular diseases, like coronary artery disease, atrial fibrillation, heart failure, systemic hypertension, etc. [36,37]. In our present study, we found increased serum levels of IL-33 in the aAT group compared to the controls, along with detectable staining for IL-33R (ST-2) in the atherosclerotic plaques, suggesting the role of the IL-33/ST-2 axis in aAT patients. IL-33 acts as an alarmin and the role of IL-33 has been reported in the activation and migration of neutrophils [38]. In recent years, the role of NETs has been widely implicated in the pathogenesis of cardiovascular diseases, including atherosclerosis [39]; however, the impact of IL-33 on neutrophil function, particularly on NET formation, has not been defined in aAT patients. Endothelial dysfunction is the hallmark feature in the development of early atherosclerotic lesions, and IL-33 has been reported to be involved in endothelial cell activation, leading to an exacerbation of inflammation [40]. The protein and mRNA levels of IL-33 and ST2 were detected in human carotid atherosclerotic plaques [40], and our study also revealed the expression of IL-33R and MPO in almost all the studied endarterectomy plaques. In the present study, we observed the significantly increased expression of IL-33R mRNA and proteins in the neutrophils of aAT compared to the controls. These observations prompted us to investigate the effect of hrIL-33 on the NET formation ability of neutrophils, and we found enhanced NETotic cells in aAT compared to the controls. We identified a unique finding that NET formation is associated with the upregulation/retention of CD16 (FCγRIII) expression on MPO+ neutrophils, as evident by the proximity of the NET filaments with CD16+ expressing neutrophils, and the significantly increased expression of the CD16 protein was observed in the MPO+ neutrophils of the aAT patients compared to the controls. We further hypothesized that CD16+MPO+ neutrophils were the major neutrophils involved in NET formation, and this is specifically regulated by IL-33 in aAT. However, this hypothesis needs further validation, and it is the subject of our future investigations.

Macrophages are one of the primary sentinels of innate immunity, and their dysfunction is extensively reported in atherosclerosis due to defective lipid metabolism [30,31,32,33]. Neutrophils are predominant leukocytes in the blood, but their role has been underestimated in the pathogenesis of atherosclerosis. In recent years, the participation of neutrophils, particularly NETs, has been widely reported in the development and progression of atherosclerosis [11,12,13]. However, the IL-33-mediated dialogue between neutrophils and macrophages was not defined in the aAT patients. After challenging the macrophages with IL-33-primed NETs, we observed the increased expression of RIPK-1, RIPK-3, and MLKL in the macrophages of the aAT patients compared to the controls, indicating the macrophage death via the activation of necroptosis pathway in the macrophages. Further, enhanced necroptosis resulted in the release of reactive oxygen species, which are considered as the driver of inflammation and cellular oxidative stress. In order to support this notion, we evaluated the status of cellular oxidative stress by measuring the levels of 8-OHdG, MDA, GSH+GSSG/GSH ratios in the isolated macrophages (challenged with IL-33-primed NETs) of the individual study subjects from each group; the levels of all these markers were significantly elevated in the patient group compared to the control group. Further, the activity of free radical scavenging enzymes was determined in the NET-treated macrophages of both groups, and the activities of both these enzymes were found to be severely compromised in the aAT patients compared to the controls. These observations revealed that the IL-33 induced the oxidative stress in the macrophages via triggering the NETs in the neutrophils. Further, in a similar experimental setup, we observed the increased transcript expression of atherogenic molecules that are highly inflammatory and may lead to the amplification of NETs and macrophage necroptosis. Increased gene expression of *NLRP-3*, *IL-1β*, and *IL-18* suggested the involvement of the IL-33-NETs axis in inflammasome activation in the macrophages. The role of MMPs was reported in plaque instability [41], and in the present study, we reported the increased expression of *MMP-3* (stromelysin-1), *MMP-9* (gelatinase B), and *MMP-12* (metalloelastase) genes via the stimulation of the macrophages with IL-33-primed NETs/neutrophils. Further, a fine balance is required between the MMPs (enzymes degrading extracellular matrix proteins) and TIMPs (biological inhibitors of MMPs) to maintain a steady-state, but we found enhanced MMPs expression and decreased *TIMP3* gene expression in the treated macrophages of aAT. These findings suggested that the IL-33-primed NETs caused the deregulation of the MMP–TIMP balance in the macrophages, and the release of more MMPs may play a vital role in plaque destabilization in aAT patients. Therefore, IL-33 appeared to be pathogenic in its function, which orchestrates the vicious cascade of potent inflammatory and atherogenic factors in the aAT patients. However, there were studies that demonstrated the protective role of IL-33 in atherosclerosis [29,39], where the development of atherosclerosis was significantly reduced in ApoE^−/−^ mice administered with IL-33 through Th1-to-Th2 switching [29]. The IL-33-mediated inhibition of atherosclerosis-associated genes, like MCP-1 and ICAM-1 (intercellular adhesion molecule 1), was also reported in the macrophages, and this anti-atherogenic action of IL-33 involved various signaling pathways, i.e., p38α, ERK1/2, JNK1/2, PI3Kγ, and NF-κB [42]. All these observations are indicative of the dual function of IL-33, i.e., as a pro- and anti-inflammatory, that may be spatiotemporal dependent in aAT.

## 5. Conclusions

In summary, the present study defined the increased expression of IL-33R on the surface of the neutrophils of aAT patients, and IL-33 played a pathogenic role in the aAT patients via the exacerbation of the NETs. We also identified a novel mechanism whereby IL-33 specifically triggered the up-regulation of CD16 expression on the neutrophils, and the CD16+MPO+ neutrophils were the dominant neutrophil populations involved in NET formation in the aAT patients. The IL-33-primed NETs have the potential to activate macrophages, thereby facilitating the NLRP3 inflammasome-mediated release of pro-inflammatory molecules that were atherogenic in their function. The elevated expression of MMP-3, MMP-9, and MMP-12 and the decreased expression of TIMP3 in the inflammatory milieu may play a critical role in plaque instability. Further, the IL-33-primed NETs induced macrophage necroptosis by enhancing cellular oxidative stress. The IL-33-driven mechanism (IL-33/ST-2/NETs-axis) eventually culminates in endothelial dysfunction and the localized amplification of inflammation in the intima of the arteries, leading to plaque destabilization. These findings were schematically represented in Figure 7. Therefore, IL-33/ST-2/NETs-axis-driven dialogue between the neutrophils and macrophages may act as a culprit in the destabilization of atherosclerotic plaques, and further investigation using a robust in vivo model is highly warranted in this direction.

## Figures and Tables

**Figure 1 antioxidants-11-02343-f001:**
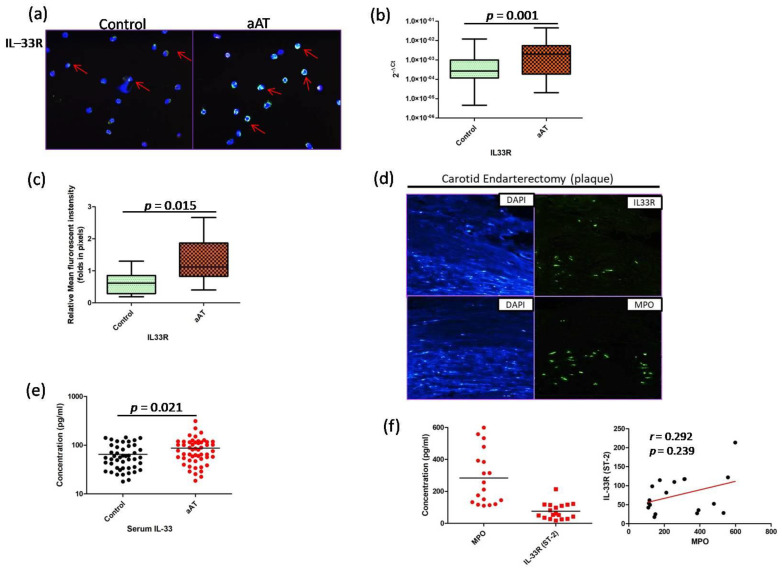
(**a**) Immunofluorescence-based expression of interleukin (IL) –33R (red arrows) in the neutrophils; (**b**) transcript expression of IL–33R in neutrophils; (**c**) quantification (relative mean fluorescent intensity) of IL–33R protein; (**d**) immunofluorescence based localization of MPO (myeloperoxidase) and IL–33R in endarterectomy plaques; (**e**) serum IL–33 levels in aAT patients and controls; (**f**) MPO and IL–33R protein quantification in plaque homogenates by ELISA, and correlation of MPO and IL–33R expression in plaques.

**Figure 2 antioxidants-11-02343-f002:**
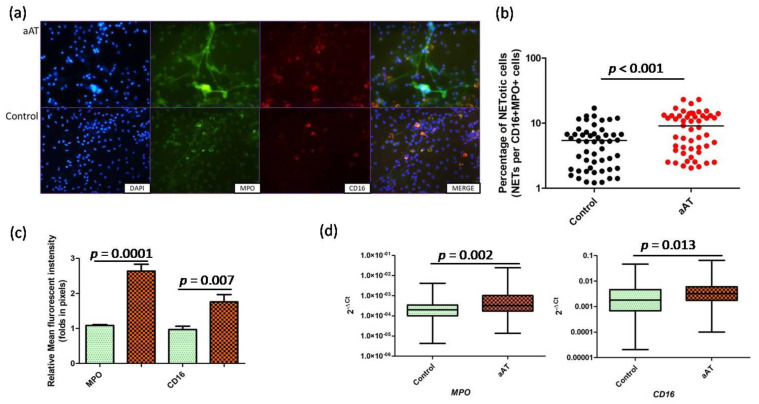
(**a**) Effect of IL-33 on in vitro neutrophil extracellular trap (NET) formation potential of MPO (myeloperoxidase) + neutrophils in aAT patients and controls (MPO = green, CD16 = red, and nucleus = blue (DAPI counterstain)); (**b**) quantification of NETotic cells (right); (**c**) quantification (relative mean fluorescent intensity) of MPO and CD16 proteins; (**d**) *MPO* and *CD16* transcript expression in IL–33–treated neutrophils of aAT patients and controls.

**Figure 3 antioxidants-11-02343-f003:**
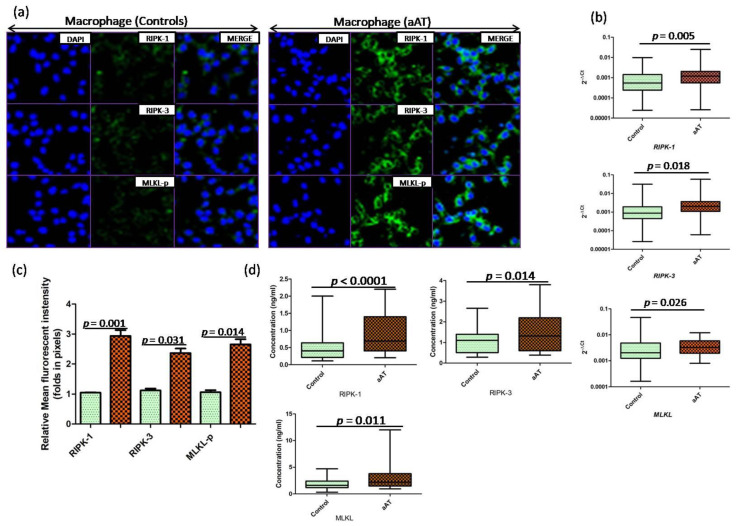
(**a**) Effect of IL–33–primed neutrophils/NETs on the necroptosis of the cultured macrophages in aAT patients and controls; (**b**) qPCR-based transcript expression of necroptosis markers (*RIPK*–*1, RIPK*–*3,* and *MLKL*); (**c**) quantification (relative mean fluorescent intensity) of necroptosis markers in the treated macrophages of the aAT patients and controls; (**d**) RIPK–1, RIPK–3, and MLKL protein quantification by ELISA in cell lysates.

**Figure 4 antioxidants-11-02343-f004:**
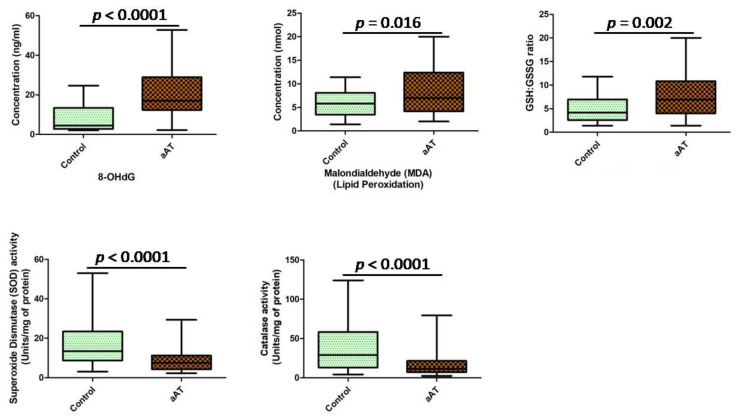
Status of oxidative stress-associated markers in the cultured macrophages after challenging with IL-33-primed neutrophils/NETs in the aAT patient and controls groups (8-OHdG = 8-hydroxy-2′-deoxyguanosine).

**Figure 5 antioxidants-11-02343-f005:**
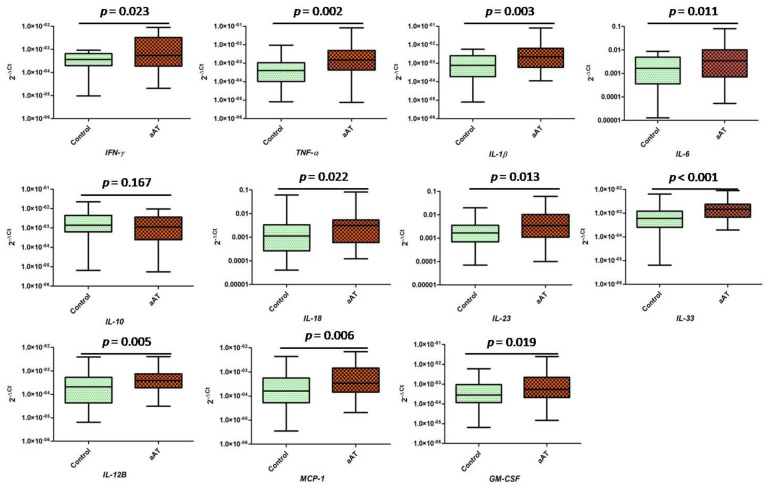
Expression (transcript) of atherogenic markers in the cultured macrophages after challenging with IL–33–primed neutrophils/NETs in the aAT patients and controls groups (IL = interleukin, *MCP =* monocyte chemoattractant protein*/CCL*–*2, GM*–*CSF* = granulocyte macrophage colony–stimulating factor).

**Figure 6 antioxidants-11-02343-f006:**
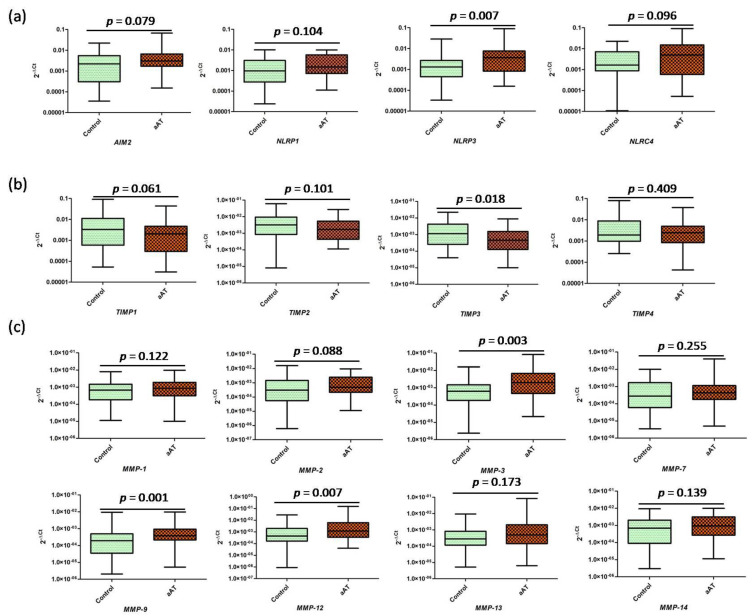
Transcript expression of (**a**) inflammsomes, (**b**) the tissue inhibitors of metalloproteinases (TIMPs), and (**c**) matrix metalloproteinases (MMPs) in the cultured macrophages after challenging with IL−33−primed neutrophils/NETs in the aAT patient and controls groups.

**Figure 7 antioxidants-11-02343-f007:**
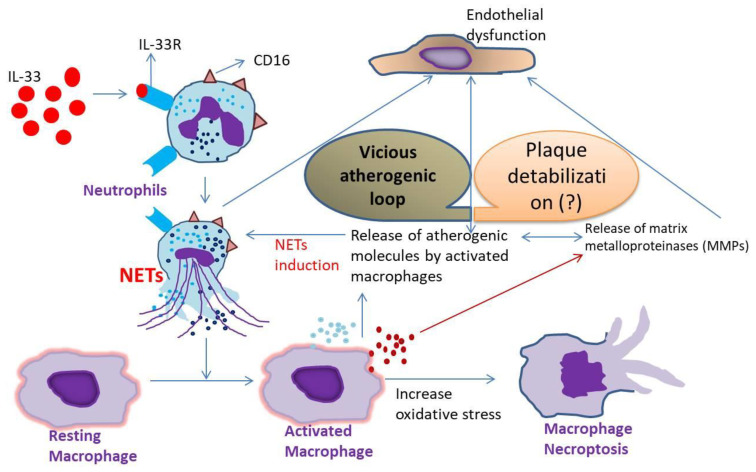
Schematic representation of the mechanism of action of IL-33 in triggering NET formation, macrophage activation and necroptosis, augmenting the release of atherogenic factors and matrix metalloproteinases (MMPs), and increased oxidative stress, collectively leading to endothelial dysfunction and the exacerbation of the disease by plaque destabilization (? = not clearly defined) (refer to the conclusion section for details).

**Table 1 antioxidants-11-02343-t001:** Demographic detail of the patient and control groups.

Clinical Parameters	Advanced Atherosclerotic Patients	Healthy Controls
Total number of subjects		
Age in years (Mean ± standard deviation (SD))	*n* = 52 (58.9 ± 21.4)	*n* = 52 (55.5 ± 22.5)
Number of Male (Age in years, Mean ± SD) =	*n* = 43 (58.6 ± 25.8)	*n* = 42 (52.7 ± 21.2)
Number of Female (Age in years, Mean ± SD) =	*n* = 9 (61.6 ± 26.3)	*n* = 10 (48.70 ± 22.4)
Hypertension	*n* = 21 (Male = 16; Female= 5)	None
Diabetes Mellitus	*n* = 16 (Male = 12; Female= 4)	None
Family history of cardiovascular diseases	*n* = 12 (Male = 10; Female= 2)	None
Percentage of carotid stenosis (>70%)	*n* = 52	None (No symptoms of angina and any other cardiovascular diseases, normal electrocardiogram)
Cholesterol (mg/dL)	123.1 ± 58.5	113.4 ± 28.2
Triglyceride	148 ± 67.3	111 ± 39.1
LDL	82.8 ± 27.6	44.2 ± 21.6
VLDL	16.8 ± 8.9	14.2 ± 7.9
HDL	35.5 ± 8.1	57.3 ± 28.5

LDL = Low-density lipoprotein, VLDL = Very-low-density lipoprotein, HDL = High-density lipoprotein.

**Table 2 antioxidants-11-02343-t002:** List of genes and their respective forward and reverse primers.

Genes	Gene Accession Number	Forward Primers (5′-3′) Reverse Primers (5′-3′)
* **β-ACTIN** *	NM_001101	GCGTGACATTAAGGAGAAG GAAGGAAGGCTGGAAGAG
* **IFN-γ** *	NM_000619	GCAGAGCCAAATTGTCTCCT ATGCTCTTCGACCTCGAAAC
* **TNF-α** *	NM_000594	CCATCAGAGGGCCTGTACCT GTGGGTGAGGAGTACATGGG
* **IL-1β** *	NM_000576	CCAAACCTCTTCGAGGCACA AGCCATCATTTCACTGGCGA
* **IL-6** *	NM_001371096	CCACCGGGAACGAAAGAGAA TCTCCTGGGGGTATTGTGGA
* **IL-10** *	NM_000572	TCTCCGAGATGCCTTCAGCAGA TAGCATCTCGGCTGGACTTCGA
* **IL-12B** *	NM_002187	TCCCTGGTTTTTCTGGCATCT CATTTCTCCAGGGGCATCCG
* **IL-18** *	NM_001386420	GCTGAAGATGATGAAAACCTGGA GAGGCCGATTTCCTTGGTCA
* **IL-23** *	NM_016584	CCCAAGGACTCAGGGACAAC TGGAGGCTGCGAAGGATTTT
* **IL-33** *	NM_033439	AATCAGGTGACGGTGTTG ACACTCCAGGATCAGTCTTG
* **IL-33R** * **(ST-2/IL1RL1)**	NM_016232	ATGTTCTGGATTGAGGCCAC GACTACATCTTCTCCAGGTAGCAT
* **GM-CSF** *	NM_000758	AATGTTTGACCTCCAGGAGCC TCTGGGTTGCACAGGAAGTTT
* **MCP-1** *	NM_002982	GACCATTGTGGCCAAGGAGA TTGGGTTTGCTTGTCCAGGT
* **MPO** *	NM_000250	TTTGACAACCTGCACGATGAC CGGTTGTGCTCCCGAAGTAA
* **RIPK1** *	NM_003804	GGAGACTAGGTGGCAGGGT CCAGTTCTGCACTCTCCAGG
* **RIPK3** *	NM_006871	TGGCCCCAGAACTGTTTGTT GGATCCCGAAGCTGTAGACG
* **MLKL** *	NM_152649	GAGGGCACTGGACAGAAACA ACTCTGCTGACTGTACCGGA
* **CD16** *	NM_000569	ATCTTCAAGCAGGGAAGCCC TGTTGCTTTGCTGTGAGGGA
* **NLRP1** *	NM_033004	GGACCAGTATCGAGAGCAGC GAGGTGAGGATGGGTCTCCT
* **NLRP3** *	NM_183395	CCTGAGCAGCCTCATCAGAA GCAAGTGCTGCAGTTTCTCC
* **NLRC4** *	NM_001199138	GAACTCGAGGCCTCACTGAA GGGCTCGGCTATTGTCCTTT
* **AIM2** *	NM_004833	TAGGTTATTTGGGCATGCTCTC ACAACTTTGGGATCAGCCTCC
* **TIMP1** *	NM_003254	GGGGACACCAGAAGTCAACC GGGTGTAGACGAACCGGATG
* **TIMP2** *	NM_003255	CAGCTTTGCTTTATCCGGGC ATGCTTAGCTGGCGTCACAT
* **TIMP3** *	NM_000362	ATGGCAAGATGTACACGGGG ATGCAGGCGTAGTGTTTGGA
* **TIMP4** *	NM_003256	CTGCCTCCCAAACCCCATTA ACATTCGCCATTTCTCCCCT
* **MMP-1** *	NM_00242	CTGTTCTGGGGTGTGGTGTC GGGCCACTATTTCTCCGCTT
* **MMP-2** *	NM_001127891	TGTGTTGTCCAGAGGCAATG ATCACTAGGCCAGCTGGTTG
* **MMP-3** *	NM_002422	AAAGACAGGCACTTTTGGCG CTTCATATGCGGCATCCACG
* **MMP-7** *	NM_002423	TACCCATTTGATGGGCCAGG AGACTGCTACCATCCGTCCA
* **MMP-9** *	NM_004994	TTCCAAACCTTTGAGGGCGA CTGTACACGCGAGTGAAGGT
* **MMP-12** *	NM_002426	AACCAACGCTTGCCAAATCC TTTCCCACGGTAGTGACAGC
* **MMP-13** *	NM_002427	GCACTTCCCACAGTGCCTAT AGTTCTTCCCTTGATGGCCG
* **MMP-14** *	NM_004995	CCGATGTGGTGTTCCAGACA TCGTATGTGGCATACTCGCC

## Data Availability

The data are contained within this article and Appendix A.

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
