# Peer review of "Interleukin-33 Induces Neutrophil Extracellular Trap (NET) Formation and Macrophage Necroptosis via Enhancing Oxidative Stress and Secretion of Proatherogenic Factors in Advanced Atherosclerosis"

_antioxidants, 2022, doi:10.3390/antiox11122343_

Round 1
Reviewer 1 Report
The manuscript is interesting but some important issues should be addressed,
In method section some methodological parts are not clear or showed.
The plaque analysis is missing. How was performed the colocalization of MPO and IL-33R in the plaques?
it is necessary to specify in the Method section that neutrophils were isolated by both study groups.
the quantification of protein in cell extracts shoulb be performed by western analysis or ELISA. the quantification by immunofluorescence is not sufficient.
Author Response
Reviewer 1
The manuscript is interesting but some important issues should be addressed,
In method section some methodological parts are not clear or showed.
- The plaque analysis is missing. How was performed the colocalization of MPO and IL-33R in the plaques?
Answer: We want to clarify that we have not performed colocalization for MPO and IL-33R in plaque samples and we have not mentioned colocalization term in the manuscript for the same. We performed the immunostaining for MPO and IL-33R by immunofluorescence to confirm the expression of these molecules in plaques. As per comments, we explained the detail method for plaque staining for MPO and IL-33R under the immunofluorescence section (briefly explained below). Changes were highlighted in red and underlined.
Plaque samples were fixed in 4% paraformaldehyde, followed by incubation in decalcifying solution (Sigma Aldrich), washed and dehydrated in increasing gradient of alcohol and embedded in paraffin. Consecutive sections (5 μm thickness) were cut and transferred on poly-L-lysine coated slides (Sigma Aldrich). Sections were deparaffinized and incubated in antigen retrieval buffer (sodium citrate, pH 6). Sections were incubated in blocking solution (Sigma Aldrich) followed by staining with optimized dilutions of primary antibodies (polyclonal goat anti-myeloperoxidase (MPO) (1:200, R & D Systems) and Rabbit polyclonal IL-33R (ST2) (1:100, Abcam) and FITC-conjugated (anti-goat and anti-rabbit) secondary antibodies (1:500 dilutions; Abcam) using a standard protocol. Counter staining was performed using 4ʹ,6-diamidino-2-phenylindole (DAPI). Sections were mounted in Vectashield mounting medium (Vector Laboratories).
(Note: Co-localization of MPO and CD16 was performed in NETs related experiment in Figure 2)
We agree that we have not performed plaque immunofluorescence analysis. To support our immunofluorescence data (in the view of comment #3), we performed ELISA using plaques homogenates that also detected the IL-33R and MPO protein. The ELISA data validated our immunofluorescence findings. We also established the correlation between IL-33R and MPO and a positive correlation was obtained but it was not statistically significant. The ELISA data is also incorporated in the revised manuscript.
- it is necessary to specify in the Method section that neutrophils were isolated by both study groups.
Answer: We mentioned in the ‘Method section’ that neutrophils were isolated from all the study subjects from both the study groups. The same has been incorporated in the revised manuscript.
- the quantification of protein in cell extracts should be performed by western analysis or ELISA. the quantification by immunofluorescence is not sufficient.
Answer: We agree that our study is lacking protein quantification data by western analysis or ELISA. However, we would like to mention that Immunofluorescence quantification is equally acceptable as we compared the relative mean fluorescent intensities (of protein expression) between patient and controls. Moreover, western blot analysis also gives relative quantification. As per reviewer’s suggestion, we performed the ELISA (gives absolute concentration) for RIPK1, RIPK3 and MLKL in the treated macrophage lysates (in available stored samples) of patient and control subjects. We also performed ELISA for MPO and IL-33R (already addressed in comment #1). The comparison of same has been included in the revised Figure 1 and 3. The ELISA methodology is explained in method section 2.6 in revised manuscript.
Reviewer 2 Report
In the current study the authors investigated whether IL-33 affects the response of neutrophils and monocytes, i.e. on NET. The authors conclude that IL-33 primes such cells and may be a promising target in preventing atherosclerosis in the future.
Major comments: The study seems to be well performed and the analysis of 52 patients in each group quite well balanced for age and sex is remarkable.
I have a couple of suggestions:
1) The authors write in the abstract that the precise mechanism of action of IL-33 in regulating neutrophil and macrophage function is not defined. This suggests to the reader that the current study will improve this knowledge. At the end of the abstract the authors claim that IL-33 may lead to disease progression. However, the reader misses a clear statement about the novel findings about the mechanisms because as mentioned in the introduction the role of IL-33 driving immune regulation and inflammation is well known in several types of disease. This needs a better description.
2) The authors state that their findings may be a promising therapeutic target in preventing the progression of atherosclerosis. However, it is not clear why this should be the case. If you want to reduce the risk of plaque rupture by targeting IL-33 this is rather to stop the final event but not to stop progression of the disease.
3) In the result section the authors describe only p-values which is not very helpful. At first, p values tell us something about the probability that you receive similar results when replicating this type of experiment. However, the n number of patients (how was this selected) was not calculated for any of the different markers under investigation. Therefore, it may be by chance. A much better way to state their findings would be to report about effect sizes (and 5% and 95% quartiles) rather than about p values. This is important as most markers show a large overlap between the two groups and only the mean is different. Therefore, please replace p-value reports by effect sizes.
4) The description of the statistic is obviously from another manuscript but does not describe the statistical procedure presented here. I.e., no comparisons between more than two groups were performed. Further, no bars and errors are given rather the box-and whisker blots. Also, was the Shapiro Wilk always significant (data not normally distributed? Otherwise, why did you use Mann Whitney U for PCR and Student’s T-Test for the other types of analysis?
5) Please state at the beginning of the discussion clearly the new mechanistic insight that you observed for IL-33. Is it simply the contribution to NET?
Author Response
Reviewer 2
In the current study the authors investigated whether IL-33 affects the response of neutrophils and monocytes, i.e. on NET. The authors conclude that IL-33 primes such cells and may be a promising target in preventing atherosclerosis in the future.
Major comments: The study seems to be well performed and the analysis of 52 patients in each group quite well balanced for age and sex is remarkable.
I have a couple of suggestions:
- The authors write in the abstract that the precise mechanism of action of IL-33 in regulating neutrophil and macrophage function is not defined. This suggests to the reader that the current study will improve this knowledge. At the end of the abstract the authors claim that IL-33 may lead to disease progression. However, the reader misses a clear statement about the novel findings about the mechanisms because as mentioned in the introduction the role of IL-33 driving immune regulation and inflammation is well known in several types of disease. This needs a better description.
Answer: We revised the abstract appropriately and also incorporated some additional description based on the comments of other reviewers. All the changes (highlighted in red and underlined) were included in the revised manuscript.
- The authors state that their findings may be a promising therapeutic target in preventing the progression of atherosclerosis. However, it is not clear why this should be the case. If you want to reduce the risk of plaque rupture by targeting IL-33 this is rather to stop the final event but not to stop progression of the disease.
Answer: We revised the statement (Last line of the Abstract) appropriately.
“Targeting IL-33/ST-2-NETs axis may be a promising therapeutic target in preventing the adverse complications of atherosclerosis”.
However, we cannot rule out the role of the reported molecule in the disease progression, as there are substantial reports available in mouse and human subjects, where some of these molecules are also associated with disease progression.
- In the result section the authors describe only p-values which is not very helpful. At first, p values tell us something about the probability that you receive similar results when replicating this type of experiment. However, the n number of patients (how was this selected) was not calculated for any of the different markers under investigation. Therefore, it may be by chance. A much better way to state their findings would be to report about effect sizes (and 5% and 95% quartiles) rather than about p values. This is important as most markers show a large overlap between the two groups and only the mean is different. Therefore, please replace p-value reports by effect sizes.
Answer: Effect size is generally applied for normal distribution (parametric data) and it is not considered appropriate for non-parametric (non-normal distributions). We consulted biostatistician and same was suggested by them. The recommendation representation of non-parametric data is by Median (minimum-maximum) and Box-plot graph truly represents the Median (minimum-maximum). However, considering the comment, we provided the effect size (Mean±SD, Cohen’s d) calculation of all the studied parameters in a table S1 (supplementary data).
- The description of the statistic is obviously from another manuscript but does not describe the statistical procedure presented here. I.e., no comparisons between more than two groups were performed. Further, no bars and errors are given rather the box-and whisker blots. Also, was the Shapiro Wilk always significant (data not normally distributed? Otherwise, why did you use Mann Whitney U for PCR and Student’s T-Test for the other types of analysis?
Answer: Considering the comments #3 and #4, we revised the ‘Statistical Analysis’ Section as mentioned below and the same was incorporated into the revised manuscript.
All statistical analyses were performed and graphs were generated using STATA software version 16 (StataCorp LP, Texas, USA) and Graph Pad Prism 5 (San Diego, USA). In the present study, the non-parametric datasets (qPCR, calorimetric assay based oxidative stress estimation and ELISA data) was analyzed using ‘Mann Whitney U’ test (represented as boxplot graph). The immunofluorescence data (except for Figure 1 (c), where ‘Mann Whitney U’ test was used) was analyzed using the student’s t test as the data was parametric data. P value of <0.05 was set as significant. We also compared the data for effect size (i.e. Mean±SD and cohen’s d) (95% confidence interval) for all the datasets and the same was presented in supplementary Table S2.
5) Please state at the beginning of the discussion clearly the new mechanistic insight that you observed for IL-33. Is it simply the contribution to NET?
Answer: As per suggestions, we revised the Discussion section and clearly defined the IL-33 mediated mechanistic insight observed in the study.
Reviewer 3 Report
This is an iteresting paper addressing an important issue on the novel role of IL33. Neutrophils and macrophages were specifically studied.
Authors conclude that the increased expression of IL-33R on the surface of neutrophils of aAT patients and IL-33 played pathogenic role in aAT patients via exacerbation of NETs. They postulate a novel mechanism IL-33 related via up-regulation of CD16 expression on neutrophils. Moreover, they report that the IL-33 primed NETs have the potential to activate macrophage with release of pro-inflammatory molecules with proatherogenic properties.
It will be of great interest to show if inflammasome is activated. Furthermore marker of plaque destabilitazion, such as TF expression and MMPs release shoud also be included to strengthen tha main findings
Author Response
Reviewer 3
This is an interesting paper addressing an important issue on the novel role of IL33. Neutrophils and macrophages were specifically studied.
Authors conclude that the increased expression of IL-33R on the surface of neutrophils of aAT patients and IL-33 played pathogenic role in aAT patients via exacerbation of NETs. They postulate a novel mechanism IL-33 related via up-regulation of CD16 expression on neutrophils. Moreover, they report that the IL-33 primed NETs have the potential to activate macrophage with release of pro-inflammatory molecules with proatherogenic properties.
It will be of great interest to show if inflammasome is activated. Furthermore marker of plaque destabilitazion, such as TF expression and MMPs release should also be included to strengthen the main findings
Answer: We studied the expression of IL-1β and IL-18 (Figure 5) that are considered as the well-known markers of inflammasome activation and are released after the activation of inflammsome. However, considering the reviewer comment we investigated the expression of common inflammasomes NLRP1, NLRP3, NLRC4 and AIM2 (Figure 6). No significant difference was found in the transcript expression of inflammasome markers except for NLRP3 expression that was found to be significantly higher in treated macrophages of patients compared to controls suggesting the NLRP3 mediated activation of inflammasome. Further, we investigated the mRNA expression levels of tissue inhibitors of metalloproteinases (TIMPs) i.e. TIMP1, TIMP2, TIMP3 and TIMP4, TIMP3 (P < 0.05) was found to be significantly decreased in the treated macrophages of patients compared to controls and similar trend was observed for TIMP1 but the difference is not statistically significant (P = 0.061) (Figure 6). We also studied the transcript expression of selective matrix metalloproteinases (MMPs) i.e. MMP-1, MMP-2, MMP-3, MMP-7, MMP-9, MMP-12, MMP-13 and MMP-14 (Figure 6). We found increased expression of MMP-3, MMP-9 and MMP-12 (P < 0.05), but no significant difference was observed for MMP-1, MMP-2, MMP-7, MMP-12 and MMP-14. All the findings were incorporated in respective method, result and discussion sections of the revised manuscript.
Round 2
Reviewer 1 Report
The paper can be accepted in this form
Reviewer 2 Report
I thank the authors for improvement of the mauscript. I have no further comments.
Reviewer 3 Report
The authors have addressed all the raised issues